# RAND: Robustness Aware Norm Decay For Quantized Seq2seq Models

## Abstract

With the rapid increase in the size of neural networks, model compression has become an important area of research. Quantization is an effective technique at decreasing the model size, memory access, and compute load of large models. Despite recent advances in quantization aware training (QAT) technique, most papers present evaluations that are focused on computer vision tasks, which have different layer composition and training dynamics compared to sequence tasks. In this paper, we first benchmark the impact of popular techniques such as straight through estimator, pseudo-quantization noise (PQN), learnable scale parameter, clipping, etc. on 4-bit seq2seq models across a suite of speech recognition datasets ranging from 1,000 hours to 1 million hours, as well as one machine translation dataset to illustrate its applicability outside of speech.

Through the experiments, we report that accuracy suffers when there is insufficient regularization signal flowing back to the outliers. We propose to construct the quantization scale as different functions of the outliers in order to regularize them as part of the end-to-end learning problem (outperforming popular learnable scale and clipping methods). PQN-QAT shows a larger improvement under the proposed method, and it opens up the possibility to exploit some of its other benefits: 1) training a single model that performs well in mixed precision mode and 2) improved generalization on long form speech recognition.

## 1 Introduction

Sequence-to-sequence (seq2seq) model is an influential class of neural architecture in various research fields and real-world AI systems. Due to the emerging demands from user-interactive devices and services (e.g., search by voice, voice assistant, etc.), end-to-end (E2E) automatic speech recognition (ASR) (Wang et al., 2019; Hannun et al., 2014; Graves, 2012; Chorowski et al., 2015; Dong et al., 2018; Li et al., 2020; He et al., 2019; Chiu et al., 2018; Kim et al., 2017; Li et al., 2019; Zeyer et al., 2020) has been widely investigated and has seen dramatic quality improvements recently. With the emergence of large models and the limited budgets on hardware, reducing the inference and training cost through model compression is a core problem for these devices and services.

Quantization reduces the number of bits required to represent weight tensors and activations, and is an effective technique at decreasing the model size, memory access, and compute load of large models. A standard approach of applying model quantization is through post-training quantization (PTQ) with int8. At int4 or lower precision, PTQ typically shows a large gap in accuracy compared to float inference (Abdolrashidi et al., 2021). Quantization aware training (QAT) (Nguyen et al., 2020; Prasad et al., 2020; Ding et al., 2022a) amends the training process with noisy operations that models the quantization errors encountered during inference, and is usually needed to close the gap.

Existing QAT techniques are generally implemented using real quantization noise or pseudo-quantization noise (PQN), which introduce training instabilities specific to their noise distribution. Injecting real quantization noise rounds the weights to integer precision during training, and requires a straight through estimator (STE) (Bengio et al., 2013) to bypass the round function during backpropagation (Courbariaux et al., 2016). STE-QAT has been observed to cause weights to not converge and oscillate around quantization decision boundaries (Nagel et al., 2022). On the other hand, PQN-QAT exposes weights to a randomly sampled noise even for weight values that would incur zero quantization rounding error. One way of reducing the weight noise variance is to introduce

extra learnable scale parameters that can be regularized with properly designed gradients (Esser et al., 2020; Baskin et al., 2021b; Jain et al., 2020; Park et al., 2022). Since the learned scale cannot be guaranteed to cover the entire range of weight values, clipping must be introduced. We denote these methods as learnable scale and clip (LSC).

In this work, we advocate for using a $L_p$ norm of each group of weight as the scale for that group during QAT (instead of extra learnable scale parameters). Groups commonly are defined as different tensors, output channels, or sub-channels. This allows for multiple outliers (i.e., scale candidates) within the channel to be simultaneously regularized in an E2E fashion, at every training iteration. Since the output of the method is a set of more quantization-friendly weights, the inference time scale still can be constructed via the maximum value with or without clipping. More specifically, this paper makes the following contributions:

1. Share benchmark results for QAT on multiple large scale speech recognition and one machine translation seq2seq models and datasets, which expands the mostly computer vision and language modeling focused QAT literature.

2. Propose robustness aware norm decay (RAND), which directly uses the weight matrix's $L_p$ norms as the quantization scales and decay those norms in an E2E QAT procedure. The performance of RAND beats LSC for a range of settings: per tensor and per channel scale, single-domain and multi-domain models and tasks.

3. Show that PQN-QAT is more sensitive to scale (i.e. the size of the hyper-rectangle that we are optimizing the loss over), as seen from the larger performance gap with and without RAND when compared to that of STE-QAT.

4. Demonstrate how the RAND enhanced PQN model improves post-training selection of layer precision to enable mixed precision inference without additional training time complexities.

5. Show that the RAND enhanced PQN model has generalization benefits on long form caption tasks beyond popular regularizations such as variational noise (VN) (Graves, 2011).

## 2 METHODS

### 2.1 PRELIMINARIES

As most of the edge devices are memory-bounded, we mainly focus on weight quantization in this work, though the ideas presented can be extended to activation quantization as well. A simple fully connected matrix multiplication can be written as $Y = WX$, where $Y \in \mathbb{R}^M$, $X \in \mathbb{R}^N$, and $W \in \mathbb{R}^{M \times N}$. Quantizing the entire weight matrix with a single scale parameter $s \in \mathbb{R}$ involves: 1) dividing by $s$ to convert from float to int range, 2) rounding to the nearest integer, 3) clipping to the integer precision lower bound $l$ and upper bound $u$. For simplicity, we focus on symmetric uniform quantization ($l = -7$ and $u = 7$ for 4-bit quantization). Multiplication with a quantized weight matrix can be modeled as

$$Y = s \cdot \left[ \text{clip}\left( \text{round}\left( \frac{W}{s} \right), l, u \right) X \right]. \tag{1}$$

To combat the effect of outliers, every output channel can have its dedicated scale. Then, matrix multiplication can be modeled as

$$Y_i = s_i \cdot \left[ \text{clip}\left( \text{round}\left( \frac{W_i}{s_i} \right), l, u \right) X \right], 1 \le i \le M, \tag{2}$$

where $s_i \in \mathbb{R}$ and denotes the $i$-th channel's scale, and $W_i$ denotes the $i$-th row of $W$. Common QAT methods optimizes the neural network by using equation 1 or equation 2 in the forward propagation, and a straight through estimator (STE) (Bengio et al., 2013) with carefully designed gradients to bypass the clip and round functions in the backpropagation. The quantized weights can be materialized during training or created during an additional post-training quantization step.

### 2.2 ROBUSTNESS AWARE NORM DECAY FOR QUANTIZATION AWARE TRAINING

It is well known from signal processing (Widrow et al., 1996) and neural compression (Agustsson & Theis, 2020) that the uniform distribution does a good job in modeling quantization noise. Since

the maximum noise introduced by the round function is $\frac{1}{2}$, which is then amplified by $s_i$, training with noise drawn from Unif $\left[-\frac{s_i}{2}, \frac{s_i}{2}\right]$ added to the weights is an effective way to emulate the QAT process. This removes the need for STE to bypass the round function, which has zero gradient almost everywhere. Training with this uniform pseudo-quantization noise (PQN) changes equation 2 into

$$Y_i = (W_i + s_i Z_i)X, 1 \le i \le M, \tag{3}$$

where $Z_i \in \mathbb{R}^M$ and every entry is drawn from from Unif $\left[-\frac{1}{2}, \frac{1}{2}\right]$. An additional benefit of PQN-QAT is the interpretability of the learning problem, which, when focusing on $W_i$, becomes

$$\min_{W_i} \mathbb{E}_{Z_i} L(f_{W_i + s_i Z_i}), \tag{4}$$

where $L(\cdot)$ denotes the training loss over the entire dataset. Essentially, equation 4 finds the optimal $W_i$ where the average loss over a hyperrectangle centered at $W_i$ with lengths of $s_i$ in all dimensions, is minimized (see Figure 1). This is in agreement with the desirable convergence to flat minima from generalization literature (Li et al., 2018; Du et al., 2022), and we explore the secondary generalization benefits offered by PQN-QAT under our proposed method in Section 5.4.

For *robustness aware norm decay (RAND)*, we advocate for eliminating the extra trainable scale parameter $s_i$ and directly computing it as a function of the network weights. In particular, set $s_i \triangleq c \left\|W_i\right\|_p$ in equation 3, where $c$ is a constant chosen for the entire neural network, and $\left\|\cdot\right\|_p$ is the vector $L_p$ norm.

$$Y_i = (W_i + c \left\|W_i\right\|_p Z_i)X, 1 \le i \le M \tag{5}$$

is the generalized equation for matrix multiplication under RAND.

## 2.3 TRAINING MODES FOR ROBUSTNESS AWARE NORM DECAY

For equation 5, we expand on three representative settings for $p$ and $c$:

$$\textbf{RAND Mode 1: } p = \infty, c = \frac{1}{2^{\text{bit}-1} - 1} \tag{6}$$

$$\textbf{RAND Mode 2: } 2 < p < \infty, 0 \le c \le \frac{1}{2^{\text{bit}-1} - 1} \tag{7}$$

$$\textbf{RAND Mode 3: } p = 2, c \ge 0. \tag{8}$$

**RAND Mode 1:** This mode sets $s_i \triangleq \frac{\max_j |W_{i,j}|}{2^{\text{bit}-1}-1}$ to fully cover the range of $W_i$ such that clipping is not needed. It converts equation 5 into

$$Y_i = \left(W_i + \frac{\max_j |W_{i,j}|}{2^{\text{bit}-1} - 1} Z_i\right) X, 1 \le i \le M. \tag{9}$$

For every channel, the term $\max_j |W_{i,j}|$ determines the maximum amount of quantization noise that the weights needs to be exposed to, and is differentiable. The loss gradient with respect to the noisy perturbation directly informs how much $\max_j |W_{i,j}|$ (i.e., the biggest outlier) should be decayed. Thus, we call the general method *robustness aware norm decay* because the uniform weight noise introduces perturbation robustness into the learned set of weights (i.e., prefer the flatter minimum in Figure 1), and simultaneously the per channel $L_p$ norm of $W$ is being decayed depending on how sensitive that output channel is to weight perturbations (i.e., decreases the width of the shaded region that the loss is averaged over in Figure 1).

**RAND Mode 2:** Mode 1 performs well when the ratio $P/Q$ is small, where $P$ is the number of parameters and $Q$ is the number of scales in a weight tensor. That does not hold for quantization with per tensor scale ($Q = 1$). In modern, large neural networks, where $P > 10^5$, that assumption may not hold even with per channel scale. Equation 9 struggles when $P/Q$ is large because only $Q$ weight entries out of $P$ candidates receive gradients from the noisy perturbation. As soon as the biggest outlier gets attenuated, it is likely that there is another one with similar magnitude at the next training step. $L_\infty$ norm decay would struggle to fully control the outliers (and thus the noise variance).

Thus, going from per tensor, to per channel, to sub-channel scale not only improves the representation (more precise dynamic range coverage), it also improves the optimization (regularizes more outliers per iteration). RAND Mode 2 expands this hierarchy by replacing the infinity norm:

$$Y_i = \left(W_i + \frac{\left\|\text{top\_k}(|W_{i,:}|)\right\|_p}{2^{\text{bit}-1} - 1} Z_i\right) X, 1 \le i \le M, \tag{10}$$

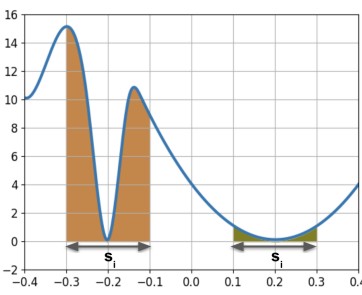

Figure 1: 1-D example of two possible training losses (shaded areas). RAND training with noise prefers the flat minimum on the right. Norm decay attenuates $s_i$ in an E2E fashion.

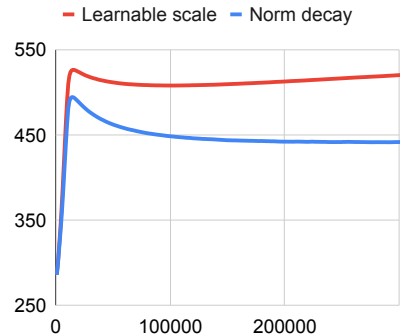

Figure 2: The $L_2$ norm of all network weights as a function of training iterations, for the ConformerS architecture on LibriSpeech.

where we constrain the number of entries contributing to the norm to $k$ to make training more stable. The general $p$-norm softens the dependency on a single max value and allows the gradients to attenuate multiple outliers per training iteration. This achieves the optimization benefits without requiring the infrastructure support needed for finer grained scale parameters. We show the improvement of Mode 2 in Sections 5.2 and 5.3. Section C contains ablation on $k$ and $p$.

**RAND Mode 3:** Equation 8 can be viewed as extensions of training with noisy weights (e.g., variational noise (Graves, 2011)). Intuitively, output channels corresponding to large weight values contain features with higher importance and are more sensitive to small input perturbations. On the contrary, as training proceeds, an output channels with increasing weight magnitudes will experience (relatively) weaker perturbations because the noise variance is held constant. Scaling the noise by the $L_2$ norm of the corresponding channel allows the noise variance to adjust with the weights. As this work is focused on QAT, we do not report empirical results for Mode 3.

## 3 RELATED WORKS

### 3.1 NEURAL NETWORK QUANTIZATION AND QUANTIZATION AWARE TRAINING

To decrease the latency and model size without compromising recognition quality, network quantization has been widely explored on ASR models (Han et al., 2015; Alvarez et al., 2016; He et al., 2019; Prasad et al., 2020; Nguyen et al., 2020; Kim et al., 2021; Ding et al., 2022a). In general, most of the existing research focus on weight and/or activation quantization. Weight quantization can save memory footprints on devices, while activation quantization can further improve computational efficiency by using integer multiplication. Among model quantization methods, post-training quantization (PTQ) with int8 is popular and easy to use for inference on edge devices. It is successfully applied in multiple applications (He et al., 2019; Sainath et al., 2020). Advanced PTQ methods (e.g., optimal brain compression) seek to exploit the correlations in the feature activations to determine the optimal quantization rounding policy (Nagel et al., 2020; Hubara et al., 2020; Frantar & Alistarh, 2022). These works are complementary to QAT and our work.

For lower-bit quantization, QAT is usually needed to mitigate the loss of precision, as shown in Nguyen et al. (2020); Prasad et al. (2020); Ding et al. (2022a). These popular QAT methods expose the network to quantization error, and use STE to bypass the round function, which has zero gradient almost everywhere. Our work seeks to improve the outlier control part of QAT to remove the need for dedicated scale parameters. This is shown to improve STE-QAT and PQN-QAT.

### 3.2 NEURAL NETWORK TRAINING WITH NOISY PERTURBATIONS

Neural networks can be trained with either noisy perturbations on the weights or features for the goal of generalization, quantization robustness, or adversarial robustness. For example, variational noise (Graves, 2011) reformulates neural network training as Bayesian inference, and gives a Bayesian

interpretation to training with Gaussian noise added to every weight. It has been widely adopted in the ASR community for improving generalization (Gulati et al., 2020; Li et al., 2021). Our work and other noise based quantization works (Baskin et al., 2021b; Défossez et al., 2022; Park et al., 2022) can be viewed as modifying the VN distribution to be specific to the uniform quantiztion noise introduced at each weight tensor, though this loosens the Bayesian connection.

One benefit of PQN-QAT over STE-QAT is that it prevents weight oscillation near the quantization decision boundary (Nagel et al., 2022). Fan et al. (2020) documents this drawback of STE and proposed to stochastically select subsets of weights to remain at full precision during every training iteration. However, the quantized weights are still trained with STE. Wang et al. (2022) augments STE-QAT with Gaussian noise to promote better convergence. Baskin et al. (2021b) (and Baskin et al. (2021a) for non-uniform quantization) proposes to use uniform noise in the QAT process, but the paper advocates to slowly progress to STE-QAT as the training proceeds. Défossez et al. (2022) uses uniform noise throughout the training, and is comparable to our work. However, the paper's exposition on the training procedure is done through an example where weights are normalized to the range of $[0, 1]$, which does not make it clear whether the weight tensor's norm can be part of the optimization to control for outliers. Additionally, Défossez et al. (2022) focuses on per tensor scale constructed from a single max value, which we benchmark to show lower performance when compared to RAND Mode 2 (see Section 5.2). To our knowledge, our work reports new findings to the field by fully analyzing the effect of end-to-end $L_p$ norm decay on PQN-QAT, and how it enables both accuracy and generalization improvements on seq2seq tasks (see Section 5.4).

Similar to QAT's goal for *weights*, adversarial robustness methods aim to train a network that is robust to *input features* that are subject to a worst case perturbation within a pre-defined $L_\infty$ ball. Classic methods for generating these adversarial examples include the fast gradient sign method (Goodfellow et al., 2014) and projected gradient descent (Madry et al., 2018). Although they are outside the scope of this paper, we remark that adversarial robustness methods may be adapted to help QAT.

### 3.3 QUANTIZATION AWARE TRAINING WITH LEARNABLE SCALE

Instead of using a $L_p$ norm as the quantization scale, many works (Baskin et al., 2021b; Jain et al., 2020; Park et al., 2022) advocate for creating extra learnable parameters as either the scale or clipping bounds. The gradients for weight entries that are clipped requires a STE, which can cause those weights to increase in magnitude even though they are already outside of the clipping bounds. Figure 2 plots the $L_2$ norm of all weights after training with learnable scale (Esser et al., 2020) and our propose method to compare the difference in behaviors. A key drawback of creating dedicated scale parameters is that they are tuned to the particular precision (i.e., 4-bit aware model do not have scales for 8-bit inference). Additionally, the highly discontinuous nature of the gradients flowing to the learnable scale (see equation 3 of Esser et al. (2020)) can introduce training instabilities and causes our multi-domain experiments to diverge (see Section 5.3). For ASR, we use empirical results to argue that RAND outperforms learnable scale and clipping (LSC) methods.

## 4 EXPERIMENTAL SETUP

### 4.1 DATASETS

We conduct experiments on three ASR datasets of different scale and one machine translation dataset to systematically evaluate the proposed approach: 1) LibriSpeech Panayotov et al. (2015) (960 hour single-domain); 2) SpeechStew Chan et al. (2021) (5,000 hour multi-domain); 3) An in-house dataset (1 million hour multi-domain); 4) WMT En-Fr (see Section A for the full setup and results). We use word error rate (WER) as the evaluation metric for all the ASR experiments.

**LibriSpeech** training set contains 960 hours of speech (460 hours "clean" speech and 500 hours "noisy" speech). The test set also consists of "clean" and "noisy" versions.

**SpeechStew** has a mix of publicly available datasets from different domains, including AMI Kraaij et al. (2005), Broadcast News, Common Voice Ardila et al. (2019), LibriSpeech Panayotov et al. (2015), Switchboard/Fisher, TED-LIUM Rousseau et al. (2012); Hernandez et al. (2018), and Wall Street Journal. Following the protocol in Chan et al. (2021), we train the models using the mixed training set, and evaluate it on each individual test set.

Table 1: The number of parameters and model sizes after encoder quantization for each backbone archiecture. For ConformerL and ConformerS, the decoder remains in float. For Cascaded Conformers, the decoder is post-training quantized to int8.

| Backbone | #Params | Size with int8 encoder | Size with int4 encoder |
|---|---|---|---|
| ConformerL | 118 Million | 138 MB | 81 MB |
| ConformerS | 10 Million | 16 MB | 12 MB |
| Cascaded Conformers | 122 Million | 123 MB | 65 MB |

Our **in-house training set** consists of over 1 million hours of English speech utterances from multiple domains, including voice assistant, voice typing, video captioning, etc. Most utterances in the training set are transcribed with a teacher model, while a small number of utterances are anonymized and hand-transcribed. We use three test sets from the voice assistant, voice typing, video captioning domains for evaluation. All of the test sets contain anonymized and hand-transcribed utterances.

## 4.2 Backbone ASR Architecture and Implementation Details

We considered two ASR backbones in our experiments. For the experiments on LibriSpeech and SpeechStew, we implemented **ConformerL** and **ComformerS** as proposed in Gulati et al. (2020) to obtain better reproducibility and fair comparisons with prior studies. For the experiments on the in-house data, we implemented the a large-medium **Cascaded Conformers** as proposed in Ding et al. (2022b) for optimal WER and latency under real world settings. The total number of parameters and model sizes after quantizations are shown in Table 1.

**ConformerL** and **ConformerS** have a frontend of 80-dimensional log Mel-filterbank energies, extracted from 25ms window and 10ms shift. The two variants have 17, 16 conformer layers, with 512, 144 dimensions, respectively. In each conformer layer, both have 32-dimensional kernel in depthwise convolutions, while ConformerL has 8 attention heads in self-attention and ConformerS has 4. The LSTM layer in the RNN-T prediction network has 640 units.

The **Cascaded Conformers** have a frontend of 128-dimensional log Mel-filterbank enegies. In addition, the 4 contiguous frames are stacked, which is then sub-sampled by a factor of 3. The model is comprised of a causal encoder and a non-causal encoder, along with separate RNN-T decoders for each encoder. The causal encoder has 9 conformer layers, where there is no self-attention in the first 3 layers. Each layer has 23-frame left context per layer and no right context to strictly prevent the model from using future inputs. The non-causal encoder has 6 conformer layers, with a cumulative 900ms of right context. All of the self-attention layers have 8 heads, and all layers use causal convolution with a kernel size of 15. Each separate RNN-T decoder consists of an 320-dimensional embedding prediction network and a 384-dimensional fully-connected joint network.

## 4.3 Training and Evaluation Details

For all models, we experiment with RAND and four other competing methods that we reproduce to the best of our knowledge (referenced in Table 2). We also adapt the norm decay part of RAND to STE-QAT and report results for it. We quantize only the encoders since the they represent the overwhelming majority of parameter for ASR models. Additionally, we do not quantize convolutional layers due to their lower parameter count relative to fully connected layers.

All models are trained on Tensor Processing Unit (TPU) v3-128 with the Adam optimizer (Kingma & Ba, 2014). ConformerS and ConformerL use a base learning rate of 5.0, multiplied by the transformer learning rate schedule (Vaswani et al., 2017) with warmup steps set to 10000, and a batch size of 2048. The Cascaded Conformer use a base learning rate of 7.5, with warmup steps set to 32000, and a batch size of 4048. LibriSpeech ConformerS models are trained to 300,000 steps and evaluated at the checkpoint with the best dev set WER. SpeechStew ConformerL and the Cascaded Conformer do not overfit on their multi-domain training data, and we evaluate at exactly 200,000 steps and 700,000 steps, respectively. All evaluations use an exponential moving average version of the network weights, computed with a decay factor of 0.9999.

Table 2: ConformerS experiments for 4-bit symmetric weight quantization with per channel scale, on LibriSpeech, reported as WER $\pm$ standard deviation.

| Reference | Eval prec. | Outlier method | QAT method | Test-clean | Test-other |
|---|---|---|---|---|---|
| N/A | Float | None | None | 2.51 | 6.15 |
| Ding et al. (2022a) | Int4 | None | 4-bit STE | 2.83±0.02 | 6.58±0.06 |
| Défossez et al. (2022) | Int4 | None | 4-bit PQN | 2.82±0.03 | 6.77±0.08 |
| Esser et al. (2020) | Int4 | LSC | 4-bit STE | 2.78±0.03 | 6.53±0.10 |
| Park et al. (2022) | Int4 | LSC | 4-bit PQN | 2.63±0.01 | 6.46±0.07 |
| RAND | Int4 | Mode 1 | 4-bit STE | 2.73±0.04 | 6.34±0.03 |
| RAND | Int4 | Mode 1 | 4-bit PQN | 2.64±0.02 | 6.30±0.04 |

## 5 RESULTS

### 5.1 SINGLE DOMAIN EVALUATION: LIBRISPEECH WITH PER CHANNEL SCALE

At int4, tensors are usually quantized with a per channel scale. $P/Q$ for ConformerS with per channel scale are on the order of 100–500, and it does not require the multiple outlier decay of RAND Mode 2. In this subsection, we report the LibriSpeech WER using RAND Mode 1, and compare against STE-QAT (Ding et al., 2022a) and PQN-QAT (Défossez et al., 2022) with no explicit outlier regularization, plus STE-QAT (Esser et al., 2020) and PQN-QAT Park et al. (2022) with LSC. Table 2 shows that under per channel scale, STE-QAT or PQN-QAT with RAND show lower WER than their LSC counterparts. Wang et al. (2022) proposes augmenting Esser et al. (2020) with Gaussian noise for better convergence. We report comparisons against it at three hyperparameter settings in Section B.

### 5.2 SINGLE DOMAIN EVALUATION: LIBRISPEECH WITH PER TENSOR SCALE

To quantify the effect of RAND Mode 2, we experiment with the different QAT techniques on a ConformerS model with per tensor scale. When each weight tensor only has a single scalar as the scale, any outlier affects the dynamic range of the entire tensor. Although this setup is believed to be sub-optimal for 4-bit and lower quantization, we will show that Mode 2 improves the WER to be very competitive with that of per channel scale, without incurring the deployment cost of implementing extra scale parameters. For Mode 2 under per tensor scale, we found that $k = 4$ and $p = 8$ work the best, and report WER under this setting. Ablation studies can be found in Section C.

Table 3 shows the effectiveness of RAND on both STE-QAT and PQN-QAT. The Test-other WER are reduced from 8.41 and 9.74 to 6.43 and 6.50, respectively. The larger improvement suggests that norm decay is especially necessary for PQN-QAT. Training with weight noise can be interpreted as sacrificing model capacity to induce model robustness to perturbations in any direction. That is a stronger robustness guarantee than STE-QAT, which only aims to be robust to the direction of the inference time quantization algorithm. For example, PQN-QAT can cause weights near zero to change signs, which is impossible under STE-QAT. Thus, it is expected that PQN-QAT needs norm decay to lower the amount of perturbation the model needs to be robust to.

With Test-clean and Test-other WERs of 2.68 and 6.32, the RAND Mode 2 performance under per tensor scale is very close to the best WERs under per channel scale (2.64 and 6.30). As described in Section 2.3, this allows us to expand the hierarchy, where RAND Mode 2 can close the optimization gap between per tensor and per channel scale.

### 5.3 MULTI-DOMAIN EVALUATION: SPEECHSTEW DATASET WITH PER CHANNEL SCALE

LibriSpeech is generally viewed as an easier dataset where models exhibit overparameterization properties. The redundancy benefits offered by overparameterization helps the network be robust to random weight perturbations. To expand the empirical analysis, we report WER results for ConformerL evaluating on the multi-domain dataset SpeechStew, which is a more diverse and difficult task. For Mode 2 under per channel scale, we found that $k = 2$ and $p = 8$ work the best, and report WER under this setting.

Table 3: ConformerS experiments for 4-bit symmetric weight quantization with per tensor scale, on LibriSpeech. The references of each technique can be found in Table 2.

| Eval precision | Outlier method | QAT method | Test-clean | Test-other |
|---|---|---|---|---|
| Float | None | None | 2.51 | 6.15 |
| Int4 | None | 4-bit STE | 3.53 | 8.41 |
| Int4 | None | 4-bit PQN | 3.94 | 9.74 |
| Int4 | LSC | 4-bit STE | 2.71 | 6.66 |
| Int4 | LSC | 4-bit PQN | 2.73 | 6.30 |
| Int4 | Mode 1 | 4-bit STE | 2.74 | 6.43 |
| Int4 | Mode 1 | 4-bit PQN | 2.71 | 6.50 |
| Int4 | Mode 2 | 4-bit PQN | 2.68 | 6.32 |

Table 4: Multi-domain ConformerL experiments for 4-bit symmetric weight quantization with per channel scale, on SpeechStew dataset. CV: Common Voice; SB: Switchboard; LS: LibriSpeech; Ted: TED-LIUM; WSJ: Wall Street Journal.

| Eval Prec | Outlier Method | QAT Method | AMI IHM | AMI SDM1 | CV | LS-Test clean | LS-Test other | SB | Ted | WSJ |
|---|---|---|---|---|---|---|---|---|---|---|
| Float | None | None | 9.19 | 23.53 | 9.89 | 2.03 | 4.32 | 8.63 | 3.98 | 1.38 |
| Int4 | LSC | 4-bit STE[*] | – | – | – | – | – | – | – | – |
| Int4 | LSC | 4-bit PQN | 9.33 | 24.31 | 10.48 | 2.17 | 4.55 | 8.87 | 4.44 | 1.58 |
| Int4 | Mode 1 | 4-bit STE | 9.22 | 24.33 | 10.18 | 2.07 | 4.38 | 8.45 | 4.45 | 1.55 |
| Int4 | Mode 1 | 4-bit PQN | 9.39 | 24.23 | 10.17 | 2.07 | 4.48 | 8.58 | 4.11 | 1.44 |
| Int4 | Mode 2 | 4-bit PQN | 9.26 | 23.97 | 10.17 | 2.08 | 4.48 | 8.55 | 4.16 | 1.37 |

[*] Model failed to converge.

In Table 4, we see the same trend where, as an outlier control method, norm decay mostly outperforms LSC across the range of tasks. Interestingly, for these larger models, STE-QAT with LSC does not converge. We hypothesize that the highly discontinuous gradients that arises due to the clipping function hurts the convergence of the learnable scale parameters (see equation 3 of Esser et al. (2020)). By contrast, RAND training remains stable for larger models. Under RAND, PQN-QAT (especially under Mode 2) outperforms STE-QAT by a small margin in terms of the average WER over all datasets, and we report the additional generalization and multi-precision benefits of RAND enabled PQN-QAT in the Section 5.4.

## 5.4 LARGE SCALE EVALUATION: IN-HOUSE DATASET WITH PER CHANNEL SCALE

In a real world production setting, training different models per domain and per platform causes scaling and maintainability challenges. Ideally, the same ASR model can be used for short form (e.g., voice assistant), medium form (e.g., voice typing), and long form (e.g., video captioning), as well as multiple hardware platforms with differing support for int4, int8, and float quantization schemes. We report WER on a real world, large scale in-house dataset. As Section 5.3 established RAND as the stronger outlier control method over LSC, we now focus on it and show how the RAND enhanced PQN-QAT model shows superior generalization, multi-domain, and multi-precision properties over STE-QAT models.

**Generalization:** Long form data tends to have higher variance in terms of background noise, pause / silence / segmentation, multiple speakers, etc. Accordingly, stronger techniques are needed to make the model generalize well to these adverse conditions.

Table 5 reports the long form WER when evaluating at 8-bit and 4-bit precision, for models trained without QAT, with variational noise (VN) using a constant Gaussian noise variance for all weights, 4-bit STE-QAT, and 4-bit PQN-QAT. Generally, 8-bit inference does not require QAT to perform on par with float models, and serves as an barometer for generalization behavior in this section. As expected, VN helps generalization and improves 8-bit WER from 17.2 to 16.2. Interestingly,

Table 5: Cascaded Conformers experiments for 4-bit QAT, evaluated at 4-bit and 8-bit, on a large scale in-house long form dataset.

| Eval precisions | Outlier method | QAT method | Long form WER |
|---|---|---|---|
| Float / Int4 | None | None | 17.2 / 27.1 |
| Int8 / Int4 | None | Tuned VN | 16.2 / 20.6 |
| Int8 / Int4 | RAND | 4-bit STE | 16.1 / 17.0 |
| Int8 / Int4 | RAND | 4-bit PQN | 15.4 / 15.6 |

Table 6: Cascaded Conformers experiments for 4-bit QAT, evaluated at 4-bit and 8-bit, on large scale in-house short form and medium form datasets.

| Eval precisions | Outlier method | QAT method | Short form WER | Medium form WER |
|---|---|---|---|---|
| Float / Int4 | None | None | 4.9 / 7.4 | 3.7 / 8.7 |
| Int8 / Int4 | None | Tuned VN | 5.0 / 5.7 | 3.8 / 5.5 |
| Int8 / Int4 | RAND | 4-bit STE | 5.0 / 5.1 | 3.7 / 3.8 |
| Int8 / Int4 | RAND | 4-bit PQN | 4.8 / 4.9 | 3.7 / 3.9 |

VN, despite not being tuned to the unique quantization noise statistics in every channel in every tensor, also improves 4-bit WER from 27.1 to 20.6. However, we see that the 4-bit PQN-QAT shows the best WER at 8-bit and 4-bit, which implies that RAND with $L_\infty$ or $L_p$ norm can be useful for generalization at 8-bit or float precision as well. We leave that to future investigations.

**Multi-domain:** Table 6 reports the 8-bit and 4-bit WER on short form and medium form utterances. Overall, the proposed RAND enabled PQN-QAT has 8-bit and 4-bit WER that are the closest to the float WER, and represents the best choice for these large-scale production datasets.

**Multi-precision:** For vanilla STE-QAT without any outlier regularization, it tunes the network to perform the best at the precision that it is trained under. Even though intuitively 8-bit quantization is easier than 4-bit, Table 7 shows that the vanilla STE-QAT model has worse WER when running as 8-bit than as 4-bit. This implies that using the 4-bit aware model checkpoint at 8-bit or mixed precision (different layers at different precision) will have a big gap compared to training the model specifically for the inference time layer precision.

Table 6 shows that the models trained using RAND have lower WER when operating in 8-bit mode and very little gap to the float model performance. This also implies that, in a mixed precision setup, increasing the number of layers running in 8-bit mode is more likely to monotonically improve WER, which is a desirable property to enable training multi-precision capable models in one shot.

# 6 CONCLUSION

In this paper, we generalize the QAT technique by writing it in terms of the weight tensor's $L_p$ norm. The explicit dependence on the $L_p$ norm provides explicit regularization within the end-to-end optimization (RAND). Through large scale experiments on multi-domain seq2seq tasks, we show that constructing the quantization scale via the $L_p$ norm formulation improves performance over existing methods that use a single parameter per scale. Additionally, RAND enabled PQN-QAT outperforms STE-QAT on a range of datasets. These improvements open up the possibility of exploiting some of the secondary benefits of RAND enabled PQN-QAT: improved generalization and multi-precision properties.

Table 7: Cascaded Conformers experiments for 4-bit STE QAT without any outlier control method, showing the degraded 8-bit WER when not using RAND.

| Eval precisions | Outlier method | QAT method | Short form WER | Medium form WER |
|---|---|---|---|---|
| Int8 / Int4 | None | 4-bit STE | 5.2 / 5.0 | 4.0 / 4.0 |

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

Table 8: BLEU scores of *Base* and *Small* models for 4-bit symmetric weight quantization with per channel scale, on English-to-French newstest2014 tests.

| Eval precision | Outlier method | QAT method | *Base* model BLEU | *Small* model BLEU |
|---|---|---|---|---|
| Float | None | None | 40.5 | 38.4 |
| Int4 | None | 4-bit STE | 39.9 | 37.6 |
| Int4 | None | 4-bit PQN | 39.7 | 37.5 |
| Int4 | RAND | 4-bit STE | 40.1 | 38.2 |
| Int4 | RAND | 4-bit PQN | 40.0 | 38.2 |

## A  MACHINE TRANSLATION EXPERIMENTS

To show that the proposed method is working on other seq2seq models and modalities, we evaluate it on a WMT 2014 English-to-French machine translation (MT) task.

### A.1  MODEL AND OPTIMIZER

We implement the *Base* Transformer model with 65M parameters reported in Vaswani et al. (2017). The *Base* Transformer model composed of encoder and decoder with N = 6 layers, model dimension $d_{model}$ = 512, hidden dimension $d_{ff} = 4 \times d_{model}$ = 2048 and number of heads $h$ = 8. We also evaluate its smaller version with model dimension $d_{model}$ = 256, hidden dimension $d_{ff} = 4 \times d_{model}$ = 1024 and number of heads $h$ = 4. This model has 30M parameters, we label it as *Small*.

Above models use SentencePiece (Kudo & Richardson, 2018) subword tokenizer with 32K vocabulary. We use the same parameters for Adam optimizer as in Vaswani et al. (2017) but with *warmup_steps* = 4000. As in Vaswani et al. (2017) we use residual dropout with value 0.1, but with label smoothing of value 0.

### A.2  TRAINING DATA AND EVALUATION

The models are trained on the WMT 2014 English-French training data. Then we select the best checkpoint using the dev data set English-French newstest2013, and report BLEU on English-French newstest2014 data set. BLEU is computed according to Post (2018) with parameters: *smooth_method*="exp"; *smooth_value*=0.0; *force*=False; *lowercase*=False; *tokenize*=intl; *use_effective_order*=False.

### A.3  HARDWARE

The MT model is trained on TPU v3-32 (TPU) for 100,000 iterations with batch size 2048. It takes nine hours to finish model training.

### A.4  EXPERIMENTAL RESULTS

We quantize MT transformer model weights of encoder, decoder and embeddings with int4. Bias is not quantized because it is negligible in comparison to the model weights. We use symmetric per-channel quantization. In table 8 we show BLEU scores of *Base* and *Small* models quantized with the different approaches and the improvements from incorporating RAND.

## B  ADDITIONAL EVALUATION: LIBRISPEECH WITH PER CHANNEL SCALE

Wang et al. (2022) proposes to augment STE-QAT with the learned step size quantization (Esser et al., 2020) with an additional Gaussian noise term. The motivation is to control the gradient in order to drive the solution to a flatter minimum. We replicate the experiments in Section 5.1 with the suggested hyperparameters of $k = 50$ and $0.2 \leq c \leq 0.4$. Table 9 contains the results, which shows Wang et al. (2022) improving over Esser et al. (2020) when $c = 0.2$, but still lagging the proposed RAND technique on ASR tasks.

Table 9: ConformerS experiments for 4-bit symmetric weight quantization with per channel scale, on LibriSpeech. $c$ is a hyperparameter specific to Wang et al. (2022).

| Reference | Eval prec. | Outlier method | QAT | $c$ | Test-clean | Test-other |
|---|---|---|---|---|---|---|
| N/A | Float | None | None | N/A | 2.51 | 6.15 |
| Esser et al. (2020) | Int4 | LSC | 4-bit STE | N/A | 2.78 | 6.53 |
| Wang et al. (2022) | Int4 | LSC | 4-bit STE | 0.2 | 2.71 | 6.49 |
| Wang et al. (2022) | Int4 | LSC | 4-bit STE | 0.3 | 2.79 | 6.35 |
| Wang et al. (2022) | Int4 | LSC | 4-bit STE | 0.4 | 2.87 | 6.52 |
| RAND | Int4 | Mode 1 | 4-bit STE | N/A | 2.73 | 6.34 |
| RAND | Int4 | Mode 1 | 4-bit PQN | N/A | 2.64 | 6.30 |

Table 10: ConformerS experiments for 4-bit symmetric weight quantization on LibriSpeech. The per tensor scale is computed via the 8-norm of the $k$ weight entries with the largest magnitude.

| $k$ | $p$ | QAT method | Test-clean | Test-other |
|---|---|---|---|---|
| 2 | 8 | 4-bit PQN | 2.77 | 6.32 |
| 4 | 8 | 4-bit PQN | **2.68** | **6.32** |
| 8 | 8 | 4-bit PQN | 2.70 | 6.41 |
| 16 | 8 | 4-bit PQN | 2.77 | 6.40 |

## C  ABLATION STUDIES FOR RAND MODE 2

For equation 10, which we replicate for convenience:

$$Y_i = \left( W_i + \frac{\|\text{top\_k}(|W_{i,:}|)\|_p}{2^{\text{bit}-1} - 1} Z_i \right) X, 1 \leq i \leq M,$$
(11)

we sweep $k$, the number of weight entries that contribute to the norm calculation, and $p$, the order of the vector norm used. This is done for the LibriSpeech per tensor experiments found in Section 5.2.

Table 10 and Table 11 show the sweep for $k$ and $p$, respectively. We conclude that $k = 4$ and $p = 8$ are closest to the optimal settings for this experiment.

Table 11: ConformerS experiments for 4-bit symmetric weight quantization on LibriSpeech. The per tensor scale is computed via the $p$-norm of the $4$ weight entries with the largest magnitude.

| $k$ | $p$ | QAT method | Test-clean | Test-other |
|---|---|---|---|---|
| 4 | 2 | 4-bit PQN | 2.73 | 6.59 |
| 4 | 4 | 4-bit PQN | 2.77 | 6.34 |
| 4 | 8 | 4-bit PQN | **2.68** | **6.32** |
| 4 | 16 | 4-bit PQN | 2.69 | 6.38 |

