# OpenReview forum: "RAND: Robustness Aware Norm Decay For Quantized Seq2seq Models"
_ICLR.cc/2024/Conference — ICLR 2024 Conference Withdrawn Submission_

### Official Review · Reviewer_Ko6F · 2023-10-29

**Soundness:** 2 fair
**Presentation:** 2 fair
**Contribution:** 2 fair
**Rating:** 3
**Confidence:** 4

**Summary:**

The paper investigates the trade-off of accurate representation between large values and small signals with Transformer models on Speech tasks. Unlike taking and optimizing explicit scaling parameters, it rewrites the quantization scheme with variational noise and proposes to adopt different Lp norms to control the trade-off.  This paper explores the relationship between weight quantization and regularized training using different Conformer models. Experimental results are presented on two Speech datasets, one in-house dataset, and a translation dataset.

**Strengths:**

* The paper builds the relationship between quantization ranges and the Lp norm in a formulated way, which is very interesting for weight quantization.
* The paper analyzes different ways that set p and c to achieve different quantization ranges.

* Experiments have been done on different datasets and different sizes of models. Especially, some tasks contain large datasets, which would cost a large effort to train them.

**Weaknesses:**

* Mode 1 looks like PQN with uniform noise and no clipping method. I feel this is like the method Relaxed quantization [1], which also views the quantization as variational noise and includes stochastic rounding as a special case. Meanwhile, I have some confusion about Mode 1 in experiment settings. What is the difference between Mode 1 + 4-bit STE and None + 4-bit STE? I expect None means no clipping method. Also, if Mode 1 + 4-bit STE means STE without any clipping, there is nothing new and is not suitable to indicate it as a good result of RAND.

* The design of Mode 2 seems empirical. It gives a strong constraint that all layers should take the same c and gives another value k in real case without sufficient explanation for doing so. Thus, Mode 2 defines quantization ranges using the norm of top-k weight ranges for all layers in the same way.


* For the implementation of LSC, how does the paper set the initialization value of scaling parameters? Another popular QAT method [2] investigates this and wins better results. It would be better if the paper could compare with a better baseline.

* The paper first claims that “For every channel, the term maxj |Wi,j | determines the maximum amount of quantization noise that the weights need to be exposed to”, which makes me think it operates in a per-channel way. However, it also claims that “Mode 1 performs well … does not hold for quantization with per tensor scale (Q = 1)”, which makes me feel confused. So, can the authors give a detailed explanation of the relationship between the three proposed modes and per-tensor, per-channel, and sub-channels?

* The writing should be improved. For example, in the contribution list, it would be better if we combine the first and the third points, and then combine the fourth and fifth points. That is to say, the first contribution becomes evaluating classic QAT methods on Speech tasks and getting some findings. The second contribution is the proposed method. The third contribution mainly introduces the achievement in the experiment part. I feel such a contribution structure can be clearer.

[1]. Relaxed Quantization for Discretized Neural Networks

[2] LSQ+: Improving low-bit quantization through learnable offsets and better initialization

**Questions:**

Please check the weakness part.

---

### Official Review · Reviewer_9JsG · 2023-11-04

**Soundness:** 3 good
**Presentation:** 3 good
**Contribution:** 1 poor
**Rating:** 3
**Confidence:** 4

**Summary:**

This paper studies model compression using quantization methods, specifically weight quantization. The authors propose RAND, a method for scalar quantization that uses the weight matrix’s norms to scale the quantization process. The authors demonstrate the efficiency of the proposed approach considering mainly speech recognition and a single machine translation experiment. Results suggest that the proposed approach is superior to the evaluated baselines.

**Strengths:**

1. The proposed method shows superior performance to the evaluated baselines on the speech recognition task.
2. The proposed approach is simple and intuitive.
3. The authors conduct experiments on both small-scale and large-scale datasets.

**Weaknesses:**

1. It seems the proposed method is a special case of previously proposed methods, which tackle the more general case.
2. The comparison to prior work can be more thorough. The authors could compare to prior work also considering other tasks and the large-scale models. Similarly, the authors could provide results for more standard benchmarks in the literature, it would be easier to compare to prior work.

**Questions:**

Overall the proposed method seems a special case of DiffQ [1] / LSQ [2], in which the authors proposed using the norm of the parameters to scale the quantization process. Additionally, the authors focus their experimental setup on speech recognition with one additional experiment on machine translation. The simplicity of the approach is not necessarily a bad thing, however, considering the specific domain of experiments, I do think the novelty of this paper is limited.

1. The authors claim that prior work on weight quantization in NN is focused on computer vision tasks with different layer compositions. However, it is not clear to me what in the proposed approach is specific to speech recognition. Additionally, from Table 2, it seems prior work performs reasonably well.

2. Both DiffQ and LSQ perform extensive evaluation of their method. Specifically, in DiffQ, the authors presented results for vision, language, and audio-related applications. However, the authors compared the proposed method considering speech recognition only. That way, it is hard to understand if the proposed method is indeed better or if the authors did not put enough effort into model optimization for the baseline methods. Can the authors provide experimental results for the tasks presented in the DiffQ paper as well?

3. The authors compared their method to baseline methods considering the speech recognition using small-scale datasets only. What is the reason the authors did not perform the same comparison for speech recognition with a large-scale dataset and machine translation?

I'm willing to change my score in case I missed something, however, considering the fact that the proposed method is not vastly different than prior work, I expect the authors to provide convincing experimental results to show the proposed method is indeed preferable.

[1] Défossez, Alexandre, Yossi Adi, and Gabriel Synnaeve. "Differentiable model compression via pseudo quantization noise." arXiv preprint arXiv:2104.09987 (2021).

[2] Esser, Steven K., et al. "Learned step size quantization." arXiv preprint arXiv:1902.08153 (2019).

---

### Official Review · Reviewer_iZJg · 2023-11-10

**Soundness:** 2 fair
**Presentation:** 2 fair
**Contribution:** 3 good
**Rating:** 5
**Confidence:** 4

**Summary:**

This paper proposes a quantization-aware training scheme for sequence-to-sequence models. During quantization of neural network weights, it is important to choose the **scale** by which float-64 or float-32 parameter values must be scaled, before obtaining a quantized representation using a low-precision format such as int8 or int4. The scalae factor corresponds to the dynamic range of the quantizer, i.e., the maximum and minimum allowable values within which the quantizer input must lie. If the quantizer input lies outside this range, the quantizer is said to be **saturated** and the input is **clipped** to the maximum or minimum value. This is, in general, undesirable, because clipping (or saturation) can lead to unbounded quantization errors, leading to instability of the training / inference process. The quantizer input values that lie outside this dynamic range (and consequently saturates the quantizer) are referred to as **outliers**.

As expected, it is quite important to choose the scale parameter (or hyper-parameter, if treated as so) appropriately, so that the effect of outliers is mitigated. The quantization scheme of this paper essentially proposed alternative options for choosing the dynamic range of the quantizer, which makes it **robust** to such outliers during quantization aware training, and hence, the title of this paper.

The authors consider specific frameworks for quantization neural network, referred to as STE-QAT (Straight-Through Estimator -- Quantization Aware Training) and PQN-QAT (Pseudo-Quantization Noise: Quantization Aware Training). In the latter strategy, the quantization noise is modeled as a uniformly distributed random variable over a bounded range, which is taken into account while computing the gradients during backpropagation. The range of this bounded random variable is proportional to the **scale** chosen for the quantizer, since larger range for a fixed number of bits implies larger quantization error. Clearly, a larger scale would increase robustness to outliers, but also decrease the utility of the trained and quantized model -- hence, the authors show via extensive numerical experiments that their proposed choice for the dynamic range is better than existing alternatives.

Please rectify if I am mistaken in my understanding of the paper. I would be more that happy to rectify it in that case.

**Strengths:**

The paper considers an important problem of quantization of neural network weight parameters, from the perspective of outlier robustness (i.e., preventing quantizer unsaturation). the proposed quantization algorithm has tunable knobs (such as the hyper-parameter $p$), which allows the user the freedom to control the extent of robustness desired.

Whats quite commendable is the extensive numerical evaluations on sequence-to-sequence models, on different tasks and different neural network architectures. They also mention the limitations of LSC and other QAT methods, and how their proposed method(s) overcomes these shortcomings.

It is quite interesting that existing benchmarks, such as *Learnable Scale and Clip (LSC)*, wherein the **scale** parameter of the quantizer is learnt jointly during the training process. Intuitively, one would expect such a joint training process to converge to the optimal scale parameter and surpass the performance of any chosen/proposed scale parameter. But LSC methods, in general, perform poorly, because of the non-differentiability of the quantization operation, which makes it difficult to evaluate gradients with respect to the scale factor, leading to training instabilities.

**Weaknesses:**

I have a few concerns with this work, and I would highly appreciate it if the authors could elaborate upon those. I would be more than happy to rectify my review and/or increase my score post the rebuttal period.

1. Is there anything specific in this paper that is concerned with sequence-to-sequence models? It seems that the general quantization aware training strategy can be extended to even other models. Please correct me if I am mistaken. And if there is anything specific with the proposed robustness-aware quantization scheme that particularly benefit seq2seq models, it would be appreciated if the authors highlighted it.

2. The proposed strategy is not analyzed much in detail. The authors did put some effort in explaining the intuitions behind them and their choice of the scale parameter, but in my opinion, it does lack some theoretical rigor. For instance, looking at the algorithm, it is not very clear what the choice of $p$ should be. Is it a hyper-parameter that needs to be tuned? The authors mention that the value $k = 2$ and $p = 8$ works best -- how did they arrive at this value? What was the set of $(k, p)$ values over which this hyper-parameter tuning was performed? Since this is quantization-aware **training**, isn't it expensive to retrain for every $(k, p)$ pair? A theoretical analysis would shed some light into this issue, and it will be highly appreciated if the authors could point something in this direction.

3. In general, the writing of the paper can be significantly improved. Several things were unclear during a first reading of the paper, such as follows:

(i) In the abstract, the authors mention "... accuracy suffers when there is insufficient regularization signal flowing back to the outliers..." It would be nice if this sentence is elaborated or completely avoided from the abstract. The notion of "outliers" isn't clear here and what does a "regularization signal" imply? It is my opinion that the abstract should be understandable to the general audience to most extent possible.

(ii) Some sentences like (page 2) "This allows for multiple outliers (i.e., scale candidates) within the channel to be simultaneously regularized in an E2E fashion, at every training iteration." -- is not very clear.

(iii) The authors mention that "uniform distribution does a good job in modeling quantization noise" -- It is important to note that this is **note true** in general, for the int8 or int4 (nearest neighbor) quantization. Since this is a strong modeling assumption on which the algorithm is heavily inspired, it would be highly appreciated if the authors make this explicitly clear. The uniform distribution assumption strictly holds true for **non-subtractively dithered quantization**. The origins go way back, see for example -- https://ieeexplore.ieee.org/abstract/document/4542554 (Lemma 1 and Lemma 2).

(iv) Several other statements will benefit from more deliberation such as "Mode 1 performs well when the ratio P=Q is small, where P is the number of parameters and Q is the number of scales in a weight tensor. That does not hold for quantization with
per tensor scale (Q = 1)." -- Is it just an empirical observation? What do you mean by number of scales -- does it mean number of rows, since each row has a scale? Please define terms like channel/sub-channel appropriately.

**Questions:**

I have some more questions, and would appreciate if authors addressed that.

1. Does RAND Mode 3 even help with quantization? Or just robustness?

2. "On the other hand, PQN-QAT exposes weights to a randomly sampled noise even for weight values that would
incur zero quantization rounding error." -- is this a goo thing or a bad thing?

3. Some works do that by doing post-training quantization of pre-trained models, and subsequently fine tuning them (eg., https://arxiv.org/abs/2305.14314). This circumvents the issue of having to train from scratch. It would be nice to mention such approaches in the related work.

4. "A key drawback of creating dedicated scale parameters is that they are tuned to the particular precision (i.e., 4-bit aware model do not have scales for 8-bit inference)." -- How does this work address this issue, could you please elaborate?

Once again, I'd be happy to reconsider my review post-rebuttal.